# SD-VLM: Spatial Measuring and Understanding with Depth-Encoded Vision-Language Models

**Pingyi Chen[1,2,3][*][†], Yujing Lou[3,4][†], Shen Cao[3], Jinhui Guo[3], Lubin Fan [3][‡],**
**Yue Wu[3], Lin Yang[2], Lizhuang Ma[4], Jieping Ye[3]**
[1]Zhejiang University, [2]Westlake University, [3]Alibaba Cloud Computing,
[4]Shanghai Jiao Tong University

## Abstract

While vision language models (VLMs) excel in 2D semantic visual understanding, their ability to quantitatively reason about 3D spatial relationships remains under-explored due to the deficiency of spatial representation ability of 2D images. In this paper, we analyze the problem hindering VLMs' spatial understanding abilities and propose SD-VLM, a novel framework that significantly enhances fundamental spatial perception abilities of VLMs through two key contributions: (1) propose Massive Spatial Measuring and Understanding (MSMU) dataset with precise spatial annotations, and (2) introduce a simple depth positional encoding method strengthening VLMs' spatial awareness. MSMU dataset includes massive quantitative spatial tasks with 700K QA pairs, 2.5M physical numerical annotations, and 10K chain-of-thought augmented samples. We have trained SD-VLM, a strong generalist VLM which shows superior quantitative spatial measuring and understanding capability. SD-VLM not only achieves state-of-the-art performance on our proposed MSMU-Bench, but also shows spatial generalization abilities on other spatial understanding benchmarks including Q-Spatial and SpatialRGPT-Bench. Extensive experiments demonstrate that SD-VLM outperforms GPT-4o and Intern-VL3-78B by $26.91\%$ and $25.56\%$ respectively on MSMU-Bench. Code and models are released at https://github.com/cpystan/SD-VLM.

## 1 Introduction

Vision-language models (VLMs) [1, 2, 3, 4, 5, 6, 7, 8, 9] have revolutionized how machines interpret and reason about visual content, achieving human-level performance on tasks like visual question answering (VQA). However, these models exhibit significant limitations in understanding 3D spatial concepts, particularly regarding quantitative spatial reasoning, e.g., absolute distances and physical dimensions. Even state-of-the-art models struggle with queries like "What is the size of the table in the image?", which reveals the gap between 2D perception and 3D quantitative spatial understanding. This limitation is particularly concerning given the increasing demand for models that can be operated effectively in real-world environments, such as robotics [10, 11], autonomous vehicles [12, 13], and augmented reality [14, 15].

Humans perceive spatial relationships based on building a 3D cognitive map of a scene in their mind. However, images, as the vision input of VLMs, are merely projections from the 3D scene to planes, losing much of the original 3D structural information. Although some works [16, 17, 18, 19, 20] explicitly use 3D scene representations as input to provide sufficient spatial information, we

---

[*]Work done during an internship at Alibaba Cloud Computing
[†]Equal contributions
[‡]Corresponding author

avoid introducing 3D structure data into current VLM architectures. One reason is that accurate point cloud data is hard to acquire in daily life. Besides, current 3D datasets suffer from uneven distribution and incomplete scene coverage compared with large-scale image datasets, limiting the training effectiveness and scalability with existing VLM frameworks. Recent studies [21, 22, 23] have incorporated depth maps as additional inputs to enhance the spatial comprehension of VLMs. However, depth maps alone are not enough to fully transform an image into a 3D structure due to the lack of camera intrinsics. The camera intrinsics are typically calibrated by geometric priors, such as using a checkerboard-like calibration board. Inspired by this, an intriguing idea is that if we provide not only a depth map but also sufficient and accurate physical measurements, VLMs could implicitly learn the "camera intrinsics", having a better understanding of 2D to 3D mapping.

Therefore, we assume that the unsatisfactory spatial ability of existing VLMs stems from two fundamental factors: (1) the scarcity of datasets featuring precise spatial quantitative annotations, and (2) the deficiency of the image inputs, preventing VLMs from fully understanding spatial context and conducting spatial reasoning.

Several studies have explored how to improve the spatial ability of VLMs by constructing spatial understanding datasets [24, 25, 26, 27, 28]. These datasets predominantly focus on basic qualitative spatial concepts, such as relative relations, which can be effectively addressed using only 2D visual cues. As a result, they fail to be competent in complex spatial reasoning. Datasets proposed in SpatialVLM [29] and SpatialRGPT [21] include both basic qualitative and quantitative spatial reasoning data. But their data construction pipeline relies on specific models, e.g., detection, segmentation, metric depth estimation, and camera calibration, instead of accurate physical annotations. Such a model-driven property may introduce systematic errors into the quantitative labels within the dataset.

Instead, we utilize 3D scene data with real-world scales to provide comprehensive metrically accurate annotations for 2D images. Hence, we propose MSMU dataset, namely Massive Spatial Measuring and Understanding dataset shown in Figure 1, a large-scale quantitative spatial reasoning dataset comprising about 25K images and 700K QA pairs (including 10K chain-of-thought samples) from 2K real 3D scenes, with 2.5M numerical annotations.

Furthermore, depth priors serve as an essential link bridging 2D visual perception and 3D scene understanding [21, 22, 23]. We comprehensively compare various ways to integrate depth information into VLMs and introduce a simple but effective approach, depth positional encoding (DPE). Building upon the success of positional embeddings in Transformer architectures [30, 31], DPE introduces the information along the third dimension ($z$-axis) orthogonal to the input image plane. It effectively upgrades the model's spatial awareness from 2D to 3D space by simply adding the depth positional embeddings on image features. Equipped with the depth positional encoding, we have trained a spatial generalist, SD-VLM, with our proposed MSMU data. Extensive experiments show that SD-VLM has a remarkable advantage on spatial tasks over image-only models or other VLMs with depth priors. Our contributions can be summarized as follows:

◇ MSMU dataset, a large-scale dataset is proposed for quantitative spatial reasoning, with 700K QA pairs with 2.5M numerical annotations, generated from real 3D scenes. A novel benchmark, MSMU-Bench, is also introduced to fully evaluate quantitative spatial reasoning capabilities of VLMs.

◇ We analyze different ways to integrate depth into VLMs and design a simple but effective depth positional encoding module, that equips VLMs with explicit spatial priors, bridging the gap between 2D perception and spatial understanding.

◇ Comprehensive experiments demonstrate that SD-VLM outperforms both previous VLMs and depth-encoded models achieving state-of-the-art performance on quantitative spatial reasoning tasks, validating the effectiveness of MSMU dataset and depth positional encoding.

## 2   Related Work

**Datasets Related to Spatial Analysis.** The boom of multi-modal datasets facilitates the rapid development of advanced VLMs [32, 33, 34, 35]. Many studies work on 2D spatial relationships and comprehensively evaluate the 2D spatial ability of VLMs [24, 25, 26, 27, 28, 36, 37, 38, 39]. Although spatial concepts are explicitly or implicitly included in these datasets, they are still insufficient for advanced and precise spatial understanding. Some works rely on complete 3D scans, which pose high

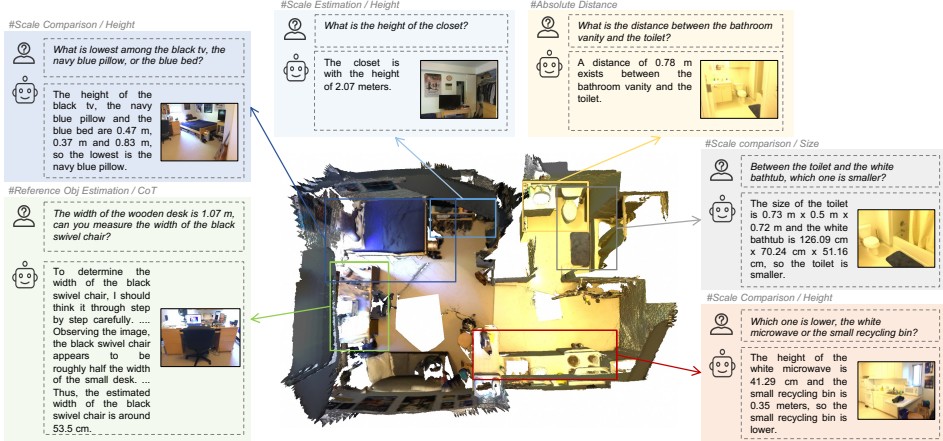

Figure 1: Demonstration of VQA pairs in MSMU. Our proposed dataset covers a range of quantitative spatial tasks involving multiple objects in the scene.

demands for integrating specific modules (e.g., a point cloud encoder) that can effectively capture 3D information [40, 41, 42, 43, 44]. Q-spatial [45] has been proposed to benchmark the quantitative spatial ability of VLMs. Although it obtains accurate manual labels, the limited data volume restricts its applicability for model training. Several datasets have tried to tackle this issue by gathering abundant images from the web. However, these datasets often rely on estimation models and lack accurate 3D annotations [29, 21].

**3D Spatial Understanding with VLMs.** Recent studies have sought to extend capabilities to 3D spatial understanding by integrating 3D information as inputs. On one hand, some works that concentrate on scene-level 3D understanding, such as scene-level captioning and scene-level visual question answering, which enable VLMs to directly process the entire scene by providing complex 3D representations like videos or point clouds [16, 17, 18, 19, 20]. On the other hand, several works discard explicit 3D inputs and use standard VLM backbones to achieve 3D spatial understanding by introducing depth information. For example, SpatialRGPT [21] incorporates the depth maps as auxiliary inputs. These depth maps are encoded and concatenated with RGB embeddings to provide additional spatial context. SpatialBot [22] is trained to adaptively call APIs to convert depth information into textual descriptions, which are then fed to the VLM as the prompt.

## 3 Problem Analysis

We review techniques for recovering 3D structures from an image. Given an image $\mathcal{I}$, there exists a mapping $\mathcal{F}$ transforming the image $\mathcal{I}$ to 3D points $\mathcal{P}$, i.e. $\mathcal{P} = \mathcal{F}(\mathcal{I})$. For any homogeneous pixel coordinates $\boldsymbol{p} = [u, v, 1]^T$ on the image, the corresponding depth value $d$ and the camera intrinsics $\boldsymbol{K}$ are required for mapping $\boldsymbol{p}$ to the 3D point $\boldsymbol{P} = [X, Y, Z]^T$:

$$\boldsymbol{P} = d \cdot \boldsymbol{K}^{-1} \boldsymbol{p}. \tag{1}$$

Therefore, an image can be mapped to its corresponding 3D points, if the depth map and camera intrinsics are available. The depth map alone is insufficient to derive a 3D point from a pixel due to missing intrinsics. The camera intrinsics could be calibrated when sufficient constraints based on the geometric priors, like using a checkerboard calibration board [46]. Theoretically, at least four line segment lengths (i.e., physical distances of corresponding 3D point pairs) in an image can be used as constraints to calibrate the intrinsics. We provide proofs in Section A.

Although 3D point clouds can be recovered from estimated depth maps and camera intrinsics [47, 48, 49, 50], we avoid explicitly integrating 3D modality into current VLMs considering the training effectiveness and scalability with existing VLM frameworks. To further avoid explicitly integrating the camera intrinsic as the input of VLMs, we tend to provide enough accurate physical measurements, enabling VLMs to implicitly learn a better mapping from 2D to 3D spatial concepts.

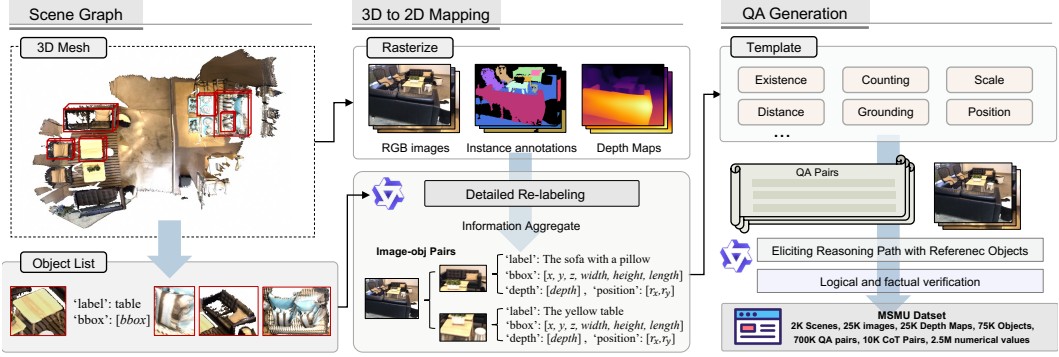

Figure 2: Overview of the data generation pipeline of MSMU. It consists of scene graph construction, 3D to 2D mapping, and QA generation.

We observe that current LLMs and MLLMs [51, 52, 53, 2, 9] demonstrate an insufficient grasp of spatial concepts like the physical sizes of objects. Their training data is not equipped with abundant spatial concepts or lacks precise numerical annotations [29, 21, 45], which motivates us to propose massive and precise spatial data.

In Section 4, we introduce the MSMU dataset with massive spatial measuring data. In Section 5, we delve into multiple approaches to fuse depth into VLMs and design a simple but effective depth-encoding strategy to integrate depth maps into VLM inputs. In Section 6, we conduct comprehensive experiments to investigate this problem.

## 4 MSMU Dataset

We elaborately design a series of spatial questions, establish a data generation pipeline to acquire spatial VQA pairs with metrically accurate annotations, and leverage LLM collaboration to generate chain-of-thought (CoT) augmented pairs.

### 4.1 Task Categories and Definitions

In contrast to previous spatial datasets [21] which mainly focus on the basic spatial understanding tasks, MSMU is designed to introduce more complex quantitative spatial tasks requiring a comprehensive and precise spatial perception ability of VLMs. MSMU contains fundamental quantitative spatial tasks, including scale estimation (e.g., height, width, or size), object grounding (2d coordinates). It also encompasses more sophisticated quantitative spatial tasks involving multiple objects or numerical outcomes, such as relative position (e.g., before/after, left/right, stand higher/lower), absolute distance measurement between two objects, scale comparison (e.g., bigger/smaller, biggest/smallest), and reference object estimation (given one scale/distance, predict the scales/distances of other objects). Furthermore, the counting task is included to assess the model's ability to discern the quantity of objects present. In addition, to eliminate the hallucination problem in VLMs, we introduce existence tasks in MSMU. In these tasks, a nonexistent object is deliberately chosen for question-answering construction. The VLM must accurately detect the absence of the object and refrain from providing misleading information. Several examples of QA pairs are shown in Figure 1.

### 4.2 Data Generation

As demonstrated in Figure 2, starting from 3D scene point clouds, we first collect the spatial information (e.g., locations, sizes, relative distances) of objects in the scene to construct a scene graph. Next, we rasterize 3D instances onto 2D images and establish a 3D-to-2D mapping, which enables transferring spatial annotations to images. We also perform filtering on both images and objects to ensure the quality of the QA pairs. Finally, we design human-verified QA templates and employ LLM collaboration to generate a rich set of QA pairs.

| Dataset | Source | Categories | | | | |
|---|---|---|---|---|---|---|
| | | Scale | Grounding | Distance | Counting | Reference Reasoning |
| SpatialBot [22] | RGBD Images | ✗ | ✗ | ✗ | ✓ | ✗ |
| Spatial-MM [27] | Images | ✗ | ✗ | ✗ | ✗ | ✗ |
| RoboSpatial [28] | 3D Scenes | ✗ | ✓ | ✗ | ✗ | ✗ |
| SpatialRGPT [21] | Images | ✓ | ✗ | ✓ | ✗ | ✗ |
| Q-Spatial [45] | Images + 3D Scenes | ✓ | ✗ | ✓ | ✗ | ✗ |
| MSMU | 3D Scenes | ✓ | ✓ | ✓ | ✓ | ✓ |

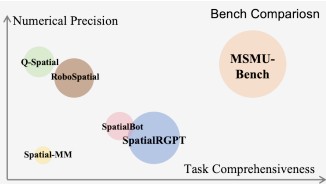

Figure 3: Comparison of different spatial datasets and benchmarks.

**Building Scene Graph.** Given a 3D point cloud of a scene, we first construct a scene graph (stored as a JSON file) to systematically organize all annotations and metadata. This graph includes the object categorization and corresponding 3D spatial localization data which provides bounding boxes for each object, defined by centroid coordinates and dimensional parameters (*width, length, height*).

**Rasterize 3D instances to 2D images.** We rasterize 3D instances onto images as masks using official tools [54]. This process bridge an object in the 3D scene and 2D image plane, making transferring spatial annotations from 3D scene graph to each image feasible.

**Image Filtering and Object Selection.** We first sparsely sample the RGB images to reduce redundancy. After that, we carefully select objects in each image, which is guided by three principal criteria: (1) Prevalence and functionality. We focus on objects demonstrating clear functional purposes which are commonly encountered in indoor environments. Architectural components (e.g., walls, ceilings) are excluded due to their limited interactive potential. (2) Instance visibility. Objects that are partially occluded (e.g., a chair mostly hidden behind a table), truncated by image borders (e.g., only a corner of a table is visible), or too small to annotate reliably (e.g., distant objects occupying fewer than 50 pixels) are excluded from our dataset. (3) Semantic disambiguation. Addressing linguistic ambiguity is important before generating annotations. For example, tables which exist in one image may vary in color or texture but are all labeled as "*table*", which brings noisy correspondence and finally misleads VLMs. To mitigate this issue, we resort to Qwen2.5-VL [8] to re-label these objects with more detailed descriptions, such as "*the white table*" or "*the wooden table*". Finally, we filter out non-informative images that have no valid objects.

**Template-based Generation.** We carefully design a set of templates based on the task definitions which include various placeholders, denoted as [·]. For instance, one template for measuring the size of a single target object is structured as follows: *"Q: What is the size of [object A]. A: The size of [object A] is [Length]×[Width]×[Height]."* For each image, we enumerate the selected objects and replace these placeholders with the corresponding object labels or spatial annotations. In tasks involving two or more target objects, we also meticulously craft instructions that incorporate all relevant object labels and spatial information. More template details are in Section B.

**Eliciting Reasoning Path with LLM collaboration.** Inspired by SpatialPrompt [45] which significantly improves the quantitative spatial ability of VLMs by eliciting reasoning paths with reference objects, we augment the QA pairs with CoT reasoning rationale via LLM collaboration. Specifically, we randomly select one object as the reference object and combine its spatial annotations along with the image as inputs to the advanced VLM, Qwen2.5-VL. The VLM is then prompted to construct a reasoning path that leverages the reference object to infer the spatial properties of another object within the image. Subsequently, we utilize a large language model, DeepSeek-V3, to assess and filter the CoT pairs by evaluating the factual consistency and logical coherence. Related prompts are provided in Section B.

**Dataset statistics.** We employ this data generation pipeline to construct VQA pairs from ScanNet [55] and ScanNet++ [54]. The resulting MSMU dataset contains 2K scenes, 25K images, 75K objects, 700K QA pairs, and 2.5M numerical values, covering a wide range of quantitative spatial tasks, as shown in Figure 3 (left). Besides, the CoT augmented group, named MSMU-CoT, consists of 10K quantitative spatial reasoning QA pairs.

### 4.3 MSMU-Bench

Existing spatial datasets struggle with annotations that lack precision, limited data volume, or insufficient task types [29, 22, 28, 45]. To address this issue, we have meticulously developed MSMU-Bench, a held-out benchmark from MSMU, designed to rigorously assess the advanced spatial reasoning capabilities of VLMs. As shown in Figure 3 (right), MSMU-Bench contains more

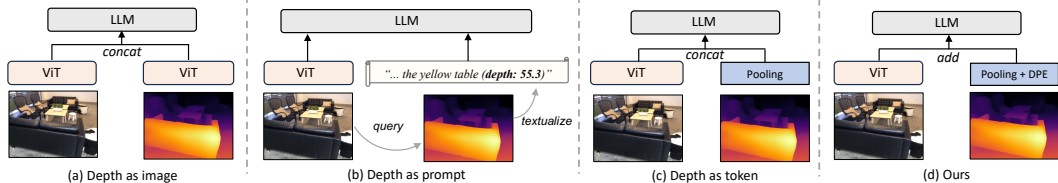

Figure 4: Illustrations of different approaches for integrating depth information.

quantitative QAs compared to other spatial benchmarks. Comprising about 1K spatial VQA pairs, this benchmark features samples from unseen scans.

We leverage GPT-4 to assess the responses generated by different models in MSMU-Bench. In the case of qualitative questions, GPT-4 assigns a score of the model's answer on a scale from 0 to 1. For quantitative queries, GPT-4 first extracts all numerical items in responses. Then, we compute the success rate by setting a threshold. For a estimated distance $\hat{d}$ and its ground truth value $d^*$, this ratio is calculated as $\delta = \max\left(\frac{\hat{d}}{d^*}, \frac{d^*}{\hat{d}}\right)$. The prompts used for scoring by GPT-4 and other details are included in Section C.

## 5 Integrating Depth into VLMs

In this section, we explore various methods to integrate depth information into VLMs, and introduce a simple but effective depth encoding method.

### 5.1 Approaches for Integrating Depth

Previous research has tried various techniques to utilize depth maps for VLMs, as shown in Figure 4. The first technique, "depth as image", was initially introduced in SpatialRGPT [21]. This method treats the depth map as a regular image and leverages a vision encoder to convert depth maps into embeddings. Besides, a learnable depth connector is required to align the depth embeddings with image embeddings. The second technique, "depth as prompt", draws inspiration from [22], of which depth values are retrieved by an API and subsequently textualized as prompts. The third technique, "depth as token", directly concatenates the depth embeddings with image embeddings.

The first method relies on the vision encoder to extract features from depth maps and introduces additional modules for training. In the second case, it is unavoidable to teach the model to utilize depth APIs. The third approach has to extend the sequence length and finally compromise the efficiency of training. Therefore, we aim to discover a straightforward integration method for incorporating depth data into VLMs, requiring minimal structural changes and training cost, while still being able to enhance spatial capabilities.

### 5.2 Depth Positional Encoding

We introduce the depth positional encoding (DPE), which can encode the depth maps into depth positional embeddings, allowing for a straightforward combination through addition.

An input depth map is represented as $\boldsymbol{D} \in \mathbb{R}^{H \times W \times 1}$. Suppose the image feature map output from CLIP is $\boldsymbol{E}^{\text{image}} \in \mathbb{R}^{H' \times W' \times d}$. We first divide the depth map into small patches $\mathcal{P}(i, j)$, matching the number of image patches. We then use adaptive mean pooling to calculate the mean depth values for each patch, and get a pooled depth map $\boldsymbol{D}' \in \mathbb{R}^{H' \times W' \times 1}$. The compact nature of these patches ensures that the average depth values provide adequately detailed depth positional information. Alternative ways of pooling depth maps are also explored in the ablation study.

Following [30], we utilize sine and cosine functions of varying frequencies to generate the depth positional embeddings $\boldsymbol{E}^{\text{depth}} \in \mathbb{R}^{H' \times W' \times d}$, which can be formulated as follows:

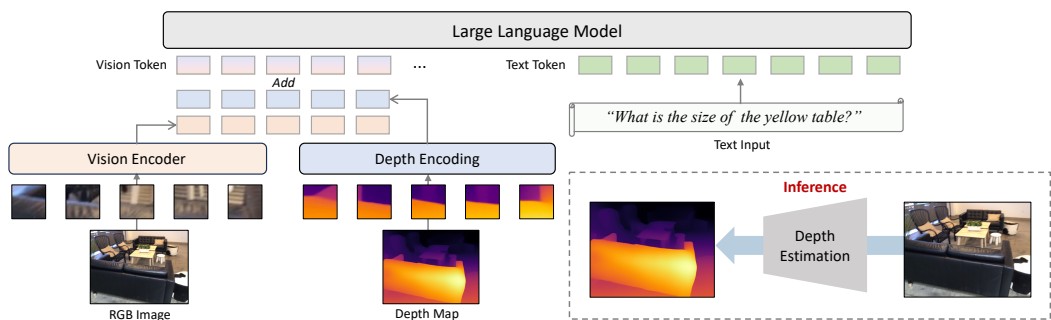

Figure 5: The architecture of SD-VLM is designed to effectively integrate spatial information into vision-language models. We incorporate an additional depth estimation module, which is particularly useful when the ground-truth depth map is unavailable.

$$\boldsymbol{E}^{\text{depth}}(i,j,2t) = \sin\left(\boldsymbol{D}'(i,j)/10000^{2t/d}\right), \quad \boldsymbol{E}^{\text{depth}}(i,j,2t+1) = \cos\left(\boldsymbol{D}'(i,j)/10000^{2t/d}\right), \tag{2}$$

where $\boldsymbol{D}'(i,j)$ denotes the depth value, $(i,j)$ represents the patch index, and $t = 0, \cdots, d/2 - 1$. Each dimension of the depth positional encoding corresponds to a sinusoidal wave.

Finally, we obtain the final vision embedding $\boldsymbol{E}^{\text{vision}}$ by integrating depth positional embeddings into image embeddings by adding, expressed as:

$$\boldsymbol{E}^{\text{vision}} = \boldsymbol{E}^{\text{image}} + \boldsymbol{E}^{\text{depth}}. \tag{3}$$

Depth-empowered vision embeddings are flattened and sent to the LLM as the final input.

The architecture of our proposed model is visualized in Figure 5. The model consists of a vision encoder to encode image features, a depth encoding module to incorporate depth information, and a large language model to process sequences of tokens. When depth maps are unavailable during inference, we employ an external depth estimation model to generate the depth map. This allows our model to adapt to various datasets and scenarios effectively.

## 6 Experiments

### 6.1 Inplementation Details

SD-VLM is built upon pretrained LLaVA-1.5-7B. The model is fine-tuned with LoRA [56] on MSMU for one epoch. The model is trained on 8 V100 GPUs, with the batch size of 2 per GPU, using 32 GPU hours. The vision encoder is CLIP-ViT/14. The external depth estimation model is Depth-Anything-V2 [48]. In the training phase, the vision encoder remains frozen. The learning rates for LLM and the projector are 2e-4 and 2e-5, respectively. The threshold for GPT-4 evaluation in MSMU-Bench is 1.25.

### 6.2 Results on MSMU-Bench

We evaluate the spatial reasoning ability of the most advanced models on MSMU-Bench. The tested models encompass a variety of proprietary VLMs, including GPT-4o [2] and Gemini-2 [3], as well as open-source alternatives that vary in model scale, such as Qwen2.5-VL [57], Intern-VL3 [9], and LLaVA-1.5-7B [4]. Additionally, we include models that incorporate depth information like SpatialRGPT [21] and SpatialBot [22]. Furthermore, we broaden our assessment to encompass LLMs, such as GPT-4-Turbo [53], Qwen2.5 [51], and DeepSeek-V3 [52]. The purpose of this evaluation is to measure these models' capacity to infer accurate responses based on common knowledge without visual inputs. The results are shown in Table 1.

Among all the baseline models, our SD-VLM stands out with the highest success rate of $56.31\%$. It excels not only in basic spatial tasks such as scale estimation, where it achieves a $51.35\%$ success rate,

| Model | Exis-tence | Object Counting | Scale Estimation | Grounding | Relative Position | Absolute Distance | Scale Comparison | Ref. Object Estimation | Average |
|---|---|---|---|---|---|---|---|---|---|
| Large Language Models (LLMs): Only Text as Input | | | | | | | | | |
| GPT-4-Turbo[53] | 12.76 | 5.21 | 13.51 | 12.64 | 24.84 | 7.50 | 36.79 | 12.04 | 15.66 |
| Qwen2.5[51] | 4.25 | 0.00 | 0.78 | 13.79 | 0.62 | 0.00 | 16.04 | 1.57 | 4.63 |
| DeepSeek-V3[52] | 0.00 | 5.24 | 1.54 | 6.90 | 10.56 | 0.00 | 25.47 | 5.24 | 7.39 |
| Vision-Language Models (VLMs): Image + Text as Input | | | | | | | | | |
| GPT-4o[2] | 44.68 | 41.67 | 3.86 | 27.59 | 67.08 | 20.00 | 54.72 | 2.09 | 32.28 |
| Gemini-2[3] | 38.30 | 43.75 | 23.94 | 19.54 | 54.66 | 12.50 | **69.81** | 18.85 | 35.17 |
| Qwen2.5-VL-72B[8] | 59.57 | 35.42 | 1.54 | 13.79 | 57.76 | 2.50 | 66.04 | 9.95 | 30.82 |
| Qwen2.5-VL-32B[8] | 29.79 | 41.67 | 10.81 | 18.39 | 60.25 | 2.50 | 46.23 | 10.99 | 27.59 |
| Qwen2.5-VL-7B[8] | 12.76 | 4.17 | 0.00 | 1.15 | 1.24 | 0.00 | 5.66 | 0.52 | 3.19 |
| Intern-VL3-78B[9] | 47.62 | 42.71 | 6.47 | 26.32 | 56.94 | 13.33 | 64.10 | 16.46 | 33.63 |
| Intern-VL3-8B[9] | 36.17 | 41.67 | 4.63 | 18.39 | 60.25 | 2.50 | 49.06 | 8.38 | 28.54 |
| LLaVA-1.5-7B[6] | 1.54 | 36.46 | 5.02 | 20.69 | 42.86 | 5.00 | 38.68 | 0.52 | 19.45 |
| Depth-Encoded Vision-Language Models : Image + Depth Map + Text as Input | | | | | | | | | |
| SpatialBot[22] | 10.64 | 46.88 | 15.83 | 28.74 | 66.46 | 5.00 | 50.94 | 8.90 | 29.17 |
| SpatialRGPT[21] | 10.64 | 36.46 | 20.08 | 17.24 | 60.25 | 15.00 | 62.26 | 9.95 | 28.98 |
| Ours | **87.23** | **47.92** | 51.35 | 42.53 | **75.16** | 40.00 | 55.66 | 46.07 | 56.31 |
| Ours w/ MSMU-CoT | **87.23** | 42.71 | **51.74** | **49.43** | 73.29 | **50.00** | **69.81** | **49.32** | **59.19** |

Table 1: Overall results of various models on MSMU-Bench. We report the results of LLMs, VLMs, and depth-encoded VLMs as a comprehensive comparison.

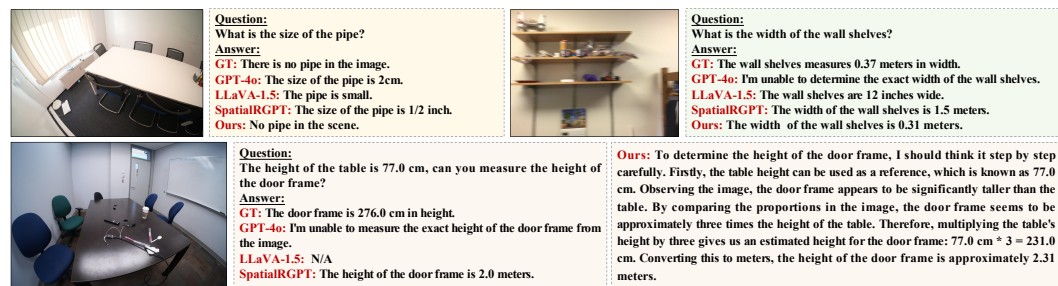

Figure 6: Examples of response from various models.

but also demonstrates significant prowess in complex spatial reasoning tasks like reference object estimation, reaching a 40% success rate compared to the second-best model's 20%. Additionally, our model's performance in the existence task is noteworthy, with the highest success rate of 87.23%, indicating its robust ability to identify the presence or absence of objects within images, with less hallucination. We also illustrate various models' responses in Figure 6.

Within the group of text-only Large Language Models, Qwen2.5, DeepSeek-V3, and GPT-4-Turbo show average success rates of 4.63%, 7.39%, and 15.66%. These results reflect the complexity of MSMU-Bench, where questions demand more than just common knowledge and benefit greatly from visual information integration. It shows the potential of our proposed MSMU dataset for improving the general spatial ability of VLMs.

**MSMU-CoT can strengthen the model's quantitative spatial ability.** We observe that employing additional CoT pairs during training can improve the spatial reasoning abilities of VLMs, particularly in complex tasks such as scale comparison, with an improvement in success rates from 55.66% to 69.81%. The average success rate also increases from 56.31% to 59.19%.

## 6.3 Results on Other Spatial Benchmarks

We have also evaluated our model on other spatial datasets including Q-Spatial++ [45] and SpatialRGPT-Bench [21]. It is important to note that SpatialRGPT-Bench refers to objects using bounding boxes or masks, which are not directly compatible with language-driven models. To address this, we have refined the benchmark by using Qwen-2.5-VL to re-annotate the objects and select the quantitative (object scales and distances) and qualitative tasks (relative positions and scale compar-

| Model | Q-Spatial++ | SRGPT-Bench | |
| --- | --- | --- | --- |
| | | Quan. | Qual. |
| GPT-4o[2] | 52.0 | 13.0 | 60.5 |
| Gemini-2[3] | 51.0 | 23.0 | 57.6 |
| Qwen2.5-VL-72B[8] | 43.6 | 16.3 | 61.4 |
| InternVL-3-78B[9] | 53.6 | 23.5 | 62.2 |
| LLaVA-1.5-7B[5] | 11.2 | 16.2 | 26.3 |
| SpatialBot[22] | 33.7 | 13.2 | 55.9 |
| SpatialRGPT[21] | 43.5 | 28.7 | 57.8 |
| Ours | **56.2** | **33.3** | **65.5** |

| Model | MSMU-Bench |
| --- | --- |
| Baseline | 46.73 |
| + depth as image | 22.64 |
| + depth as prompt | 48.78 |
| + depth as token | 35.72 |
| + DPE w/ estimated depth | 55.35 |
| + DPE-learnable | 56.18 |
| + DPE-sincos | **56.31** |

Table 2: Left: Results on other spatial datasets (Q-Spatial++ and SpatialRGPT-Bench). Right: Comparison of models with various approaches of incorporating depth priors.

| Model | MSMU-Bench | Q-Spatial++ | SpatialRGPT-Bench (Quan.) | SpatialRGPT-Bench (Qual.) |
| --- | --- | --- | --- | --- |
| LLaVA-1.5-7B | 17.3 | 11.2 | 10.3 | 26.3 |
| LLaVA-1.5-7B + DPE | **18.8** | **23.0** | **18.4** | **27.5** |

Table 3: Overall results of models which are fine-tuned with LLaVA-1.5-mix665k and tested on spatial datasets.

isons) for evaluation in SpatialRGPT-Bench. We follow the official setting when evaluating these benchmarks. As shown in Table 2 (left), SD-VLM achieves a 56.2% success rate on Q-Spatial++, surpassing all the other baselines. SD-VLM also achieves the state-of-the-art performance on the quantitative (33.3% success rate) and qualitative tasks (65.5% success rate) of SpatialRGPT-Bench which contains many outdoor scenes, revealing the strong generalization capabilities of our model. More details are included in Section D.

## 6.4 Comparison between Different Depth Integrations

We have examined various methods for incorporating depth information into our models, as shown in Table 2 (right). The result of "depth as image" suggests that treating depth maps as images may not be an effective way to integrate depth data. The subpar performance of "depth as token" suggests that adding extra tokens not only increases training costs but also makes it more challenging for VLMs to learn the interactions between depth maps and image embeddings. The approach of "depth as prompt" achieves a slight improvement from 46.73% to 47.78%. Our proposed depth positional encoding demonstrates a marked advantage, achieving a success rate of 56.18%.

An alternative way of encoding depth maps involves utilizing learnable layers that adaptively condense the depth map and convert it into depth positional embeddings. As shown in the bottom of Table 2 (right), the success rate of 56.18% is comparable to the performance of sinusoidal depth positional encoding.

We have trained our model with estimated depth maps from Depth-Anything-V2. Although its performance is not as good as the one with ground-truth depth maps due to the relatively noisy depth, it still shows a significant superiority over other depth integration methods. More experimental results on depth are included in Section E and F.

**Further investigation of depth encoding.** We have conducted further investigations into the efficacy of our proposed depth positional encoding technique. We resort to general visual instruction datasets, LLaVA-1.5-mix665k [5], to fine-tune LLaVA-1.5-7B with DPE in its instruction following training stage. This dataset, which is not explicitly equipped with spatial knowledge, allows us to examine the extent to which depth positional embeddings can enhance a VLM's spatial reasoning ability implicitly without massive spatial data. The results are demonstrated in Table 3. Our observations indicate that while models trained on general datasets may not excel on spatial tasks, incorporating depth encoding can still enhance their performance across all three spatial benchmarks, with a notable 25% relative improvement in model accuracy, which reflects the potential of depth positional encoding in eliciting the model's spatial reasoning ability.

| Depth Estimator | MSMU-Bench | Q-Spatial++ | SpatialRGPT-Bench | Average |
|---|---|---|---|---|
| DepthAnything | 56.3 | 56.2 | 33.3 | 48.6 |
| UniDepth | 56.2 | 54.7 | 32.0 | 47.6 |

Table 4: Overall results of different depth estimators.

| No Noise | $\delta = 0.1$ | $\delta = 0.3$ | $\delta = 0.5$ | $\delta = 0.7$ | No Depth |
|---|---|---|---|---|---|
| 56.3 | 55.1(-1.2) | 54.0(-2.3) | 53.3(-3.0) | 51.4(-4.9) | 46.7(-9.6) |

Table 5: Model performance with various depth noise.

## 6.5 Robustness of DPE

**Variations of depth estimators.** We have conducted an ablation study on the depth estimation backbone by replacing DepthAnything with another powerful depth estimator, UniDepth [49]. As shown in Table 4, our model maintains competitive performance across spatial benchmarks. On MSMU-Bench, the scores are nearly identical (56.3% vs. 56.2%). The average performance decreases slightly from 48.6% to 47.6% when switching to UniDepth. The results indicates that our model has learned generalizable depth priors rather than overfitting to any specific depth-estimation architecture.

**Depth noise.** We have conducted further ablation studies to evaluate the robustness of our model against depth estimation noise on the MSMU-Bench as shown in Table 5. Specifically, we inject zero-mean Gaussian noise with different standard deviations into the normalized depth maps. As shown in Table 5, as the noise level $\delta$ increases, the performance gradually declines from 56.3% under no noise to 51.4%. Despite this degradation, the model maintains competitive performance even under significant noise conditions, outperforming the setting without depth input by a clear margin. These results indicate that our DPE exhibits strong robustness to depth perturbations, effectively leveraging depth priors.

## 6.6 Performance on General Benchmarks

In this section, we evaluate whether incorporating spatial VQA data and depth information affects the model's performance on general VQA benchmarks. To ensure a fair comparison, we incorporate the LLaVA-1.5-mix665k dataset into our training data. As shown in Table 6, our model still achieves superior performance on MSMU-Bench (55.8% vs. 19.5%). Regarding other benchmarks, our model shows comparable performance to the baseline and even exhibits a slight edge on Whatsup, GQA, and VQA-v2. The overall competitive results confirm that our approach can significantly enhance spatial understanding abilities of VLMs without compromising general capabilities.

| | MSMU-Bench | Whatsup[26] | GQA[58] | TextVQA[59] | VQA-v2[60] | Vizwiz[61] |
|---|---|---|---|---|---|---|
| LLaVA-1.5-7B | 19.5 | 58.3 | 61.9 | **58.2** | 78.5 | **50.1** |
| Ours | **55.8** | **60.9** | **62.9** | 57.5 | **79.1** | 49.2 |

Table 6: Comparison of SD-VLM and base model performance on general benchmarks.

## 7 Conclusion

In this work, we identified a critical gap in the ability of Vision-Language Models (VLMs) to perform quantitative spatial reasoning. To address this, we developed MSMU, a large-scale dataset comprising 700K QA pairs and 2.5M numerical physical annotations derived from real 3D scenes, designed to provide precise metric supervision for enhancing VLMs' spatial reasoning capabilities. We introduced a simple but effective depth positional encoding module that integrates the third-dimension information into the VLM architectures, effectively upgrading the model's spatial awareness from 2D to 3D. This innovation was shown to significantly enhance spatial reasoning abilities, outperforming both RGB-only VLMs and depth-encoded VLMs. We anticipate that our contributions will pave the way for further advancements in VLMs' spatial reasoning capabilities, enabling more effective application in real-world environments.

**Acknowledgement.** Pingyi Chen was supported by Alibaba Research Intern Program. Yujing Lou was supported by Alibaba Postdoc Program. We thank Prof. Xiaoguang Han, director of the GAP Lab (Chinese University of Hong Kong, Shenzhen) and the lab members for their assistance with the dataset.

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

# A  Proof of Mapping from 2D to 3D with Distance Constraints

In this section, we prove that given an image with its depth map and enough annotated physical lengths, the mapping from the image to its 3D structure can be established.

**Mapping from a 2D image to 3D points.**  Given an image $\mathcal{I}$, there exists a mapping $\mathcal{F}$ transforming the image $\mathcal{I}$ to 3D points $\mathcal{P}$, i.e. $\mathcal{P} = \mathcal{F}(\mathcal{I})$. For any homogeneous pixel coordinates $\boldsymbol{p} = [u, v, 1]^T$ on the image, the corresponding metric depth value $d$ and the camera intrinsics are required for mapping $\boldsymbol{p}$ to the 3D point $\boldsymbol{P} = [X, Y, Z]^T$:

$$\boldsymbol{P} = \begin{bmatrix} X \\ Y \\ Z \end{bmatrix} = d \cdot \boldsymbol{K}^{-1}\boldsymbol{p} = \begin{bmatrix} (u - c_x)\frac{d}{f_x} \\ (v - c_y)\frac{d}{f_y} \\ d \end{bmatrix}. \tag{4}$$

$\boldsymbol{K}$ is the camera intrinsic matrix with four unknown parameters,

$$\boldsymbol{K} = \begin{bmatrix} f_x & 0 & c_x \\ 0 & f_y & c_y \\ 0 & 0 & 1 \end{bmatrix}, \tag{5}$$

**Constraints Based on Metric Distances.**  Camera intrinsics are necessary for mapping depth map to 3D structure.  3D point coordinates in a camera coordinate system are hard to obtain in daily scenarios, while measuring metric distances between two points is feasible.

Suppose two pixels $\boldsymbol{p}_1$ and $\boldsymbol{p}_2$ are the endpoints of a line segment with physical length $L$ in the image $\mathcal{I}$. The corresponding depth values are $d_1$ and $d_2$ and 3D points from the mapping are $\boldsymbol{P}_1$ and $\boldsymbol{P}_2$. Hence, the physical length is calculated by

$$L^2 = \|\boldsymbol{P}_1 - \boldsymbol{P}_2\|_2^2. \tag{6}$$

Explicitly, the constraint based on the metric distance is

$$L^2 = (\frac{u_1 - c_x}{f_x}d_1 - \frac{u_2 - c_x}{f_x}d_2)^2 + (\frac{v_1 - c_y}{f_y}d_1 - \frac{v_2 - c_y}{f_y}d_2)^2 + (d_1 - d_2)^2. \tag{7}$$

This is one nonlinear equation for four unknowns $f_x$, $f_y$, $c_x$, $c_y$. Suppose there are $N$ line segments labeled with physical lengths in an image. We define

$$E_i(f_x, f_y, c_x, c_y) = \|\boldsymbol{P}_{i1} - \boldsymbol{P}_{i2}\|_2^2 = L_i^2, \ \ i = 1, \cdots, N. \tag{8}$$

The residual vector is

$$r = [E_1, \cdots, E_N]^T. \tag{9}$$

Hence, the intrinsic parameters are estimated by optimization algorithms with the objective:

$$\min_{f_x, f_y, c_x, c_y} \|r\|^2. \tag{10}$$

Solving above optimization problem needs at least four segments ($N \geq 4$) with ground truth length. Actually, the depth values are sometime noisy and the ground-truth depth map is difficult to acquire in daily scenarios. Besides, if the given depth map is relative, i.e. $d = a \cdot d_{rel} + b$, there exists two more parameters. Empirically, abundant constraints would produce a more robust estimation, which improves robustness to noise and improve stability.

Theoretically, given an image with its depth map, we can fully mapping pixels to corresponding 3D point cloud when enough annotated physical lengths are provided. We believe that providing enough physical labeled lengths in images would facilitate the spatial understanding of images.

# B  Details for Data Generation

In this section, we provide more details for the data generation procedure.

### B.1 Prompt Details

**Prompts for semantic disambiguation.** We crop instances from an image, which are fed to Qwen-2.5-VL with the prompt as below:

---

Describe the object class in the image and directly return a term. For example, the red car, the wooden table, the man in white.

Output:

---

**Prompts for CoT data generation.** To elicit reasoning paths with reference objects, we randomly select an object as the reference object, combining its spatial annotations and the image as inputs to Qwen2.5-VL with the prompt as below:

---

Please help me rephrase the following VQA (Visual Question Answering) pairs to improve their rationale. I will give you an image which shows an indoor environment and contains various objects. Based on the image, I will also give you a question and answer, the question containing a reference object. You need to propose a robust step-by-step plan to answer the question by using the reference scales and the information from the image.

For example:
Q: The height of the chair is 0.7 m, can you measure the height of the wooden table?
A: Since the height of the chair is 0.7 m, I think the height of the wooden table is 1.4 m

Example Output:
To determine the height of the table. I should think it step by step carefully. Firstly, the chair height can be used as reference, which is known as 0.7 m. The wooden table appears to be about double the height of the counter. So, the height of the wooden table is 1.4 m.

Please process the following VQA pairs in the same way:
Q: [Q]
A: [A].

Output:

---

**Prompts for CoT data quality assessment.** We employ a large language model, DeepSeek-V3, to evaluate and filter the CoT pairs based on their factual accuracy and logical coherence. The relevant prompts are provided below:

Please help me evaluate the factual consistency and logical coherence between the original VQA pairs and the generated ones. The goal is to ensure that the generated answers align with the original facts and maintain logical reasoning.

Task:
Compare the original VQA pair with the generated one.
Check for factual consistency: Ensure that the generated answer does not contradict the original facts.
Check for logical coherence: Ensure that the generated answer and its reasoning (if provided) are logically sound and aligned with the original context.
Finally give a score between 0 and 10, where 0 indicates a poor match and 10 indicates a perfect match.

Input:
Original VQA Pair:
Q: [Original Question]
A: [Original Answer]
Generated VQA Pair:
Q: [Generated Question]
A: [Generated Answer]
The output should follow the format:
Factual Consistency: [Yes/No]
Logical Coherence: [Yes/No]
Score: [Score]. An example of output is
Factual Consistency: Yes,
Logical Coherence: Yes,
Score: 10.

Now return your output:

## B.2 Statistics of MSMU

We categorize the spatial tasks in MSMU into 8 types, the distribution of which is illustrated in Figure 7 (left). The QA distribution of MSMU-Bench is also shown in Figure 7 which provides a detailed breakdown of these eight categories.

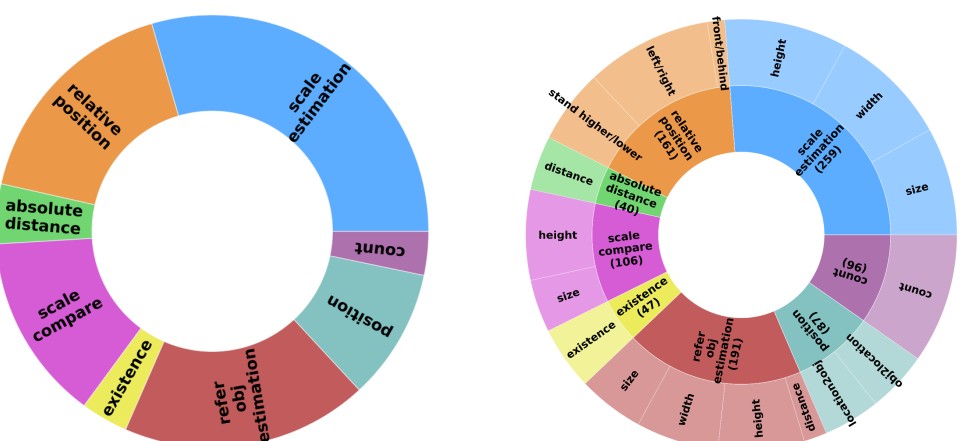

Figure 7: The left shows the QA distribution of the MSMU dataset. The right shows the QA distribution of MSMU-Bench and the specific numbers of each category.

## B.3  QA Templates

We provide templates that are used to construct the spatial tasks. Since MSMU consists of eight spatial tasks, the templates can also be grouped into eight types, which are demonstrated as below:

**Scale Estimation**

```
# Scale Estimation
size_template_questions = [ "What is the size of [A]?", "How big is [A
    ]?", "Can you provide the size measurement of [A] ?", ]
size_template_answers = [ "The size of [A] is [Length] x [Width] x [
    Height]. ", "[A] is with the length of [Length], width of [Width],
     and height of [Height]."]

height_template_questions = [ "What is the height of [A]?", "How tall
    is [A]?", "Can you measure the height of [A] ?", ]
height_template_answers = [ "The height of [A] is [Height].", "[A] is
    with the height of [Height].","[A] measures [Height] in height."]

width_template_questions = [ "What is the width of [A]?", "Determine
    the width of [A].", "Can you measure the width of [A] ?", ]
width_template_answers = [ "The width  of [A] is [Width].", "[A] is
    with the width of [Width].","[A] measures [Width] in width."]
```

**Counting**

```
# Counting
count_template_questions = [ "How many [A]s are there in the image ?",
     "what's the total number of [A]s in the image?" ]
count_template_answers = [ "There are [X] [A]s.",  "There are [X] [A]s
    in the image.", "[X]."]
```

**Grounding**

```
# Grounding
position1_template_questions = [ "What object is located at ([x],[y])?
    ", "What can you find at ([x],[y])?", "What object does the
    position ([x],[y]) belong to?", ]
position1_template_answers = [ "It is [A].", "That is [A].",'[A].']
position2_template_questions = [ "What is the coordinate of [A] ?", ]
position2_template_answers = [ "([x],[y])." ,"It is located at ([x],[y
    ]) in the image."]
```

**Existence**

```
# Existence
zero_template_questions=[ "What is the size of [A]?", "How big is [A]?
    ", "Can you provide the size measurement of [A] ?", "What is the
    height of [A]?", "How tall is [A]?", "Where is [A]?",
"How many [A]s are there in the image ?", "what's the total number of
    [A]s in the image?"]
zero_template_answers=['There is no [A] in the image','Can not find [A
    ].','No [A] in the scene.']
```

**Absolute Distance**

```
# Absolute Distance
distance_template_questions = [ "What is the distance between [A] and
    [B]?", "How far away is [A] from [B]?", "Can you provide the
    distance measurement between [A] and [B]?", ]
distance_template_answers = [ "[A] and [B] are [X] apart.", "A
    distance of [X] exists between [A] and [B].", "[A] and [B] are [X]
     apart from each other.","The distance is [X]." ]
```

### Relative Position

```python
# Relative Position
left_template_questions = [ "Is [A] to the left/right of [B] from the
    viewer's perspective?", "Does [A] appears on the left/right side
    of [B]?", "Can you confirm if [A] is positioned to the left/right
    of [B]?", ]
left_template_answers = ["Yes, [A] is to the left/right of [B].","
    Indeed, [A] is positioned on the left/right side of [B]."]

closer_template_questions=["From the viewer's perspective, what is
    closer, [A] or [B] ?"]
closer_template_answers=["[X] is more closer."]

stands_template_questions=["Which stands higher/lower in the image, [A
    ] or [B] ?"]
stands_template_answers= ["[X] stands higher/lower."]
```

### Scale Comparison

```python
# Scale Comparison
taller_template_questions=["Between [A] and [B], which one is taller/
    lower?","Which one is taller/lower, [A] or [B]? "]
taller_template_answers=["The height of [A] is [Height A] and [B] is [
    Height B], so [X] is taller/lower."]

tallest_template_questions = ["What is tallest/lowest among [A], [B],
    and [C]?"]
tallest_template_answers = ["The  height of [A] is [Height A], height
    of [B] is [Height B], and height of [C] is [Height C], so the
    tallest is [X]."]

larger_template_questions=["Between [A] and [B], which one is larger/
    smaller?","Which one is larger/smaller, [A] or [B]? "]
larger_template_answers=["The size of [A] is [Length A] x [Width A] x
    [Height A] and [B] is [Length B] x [Width B] x [Height B], so [X]
    is larger/smaller."]
```

### Reference Object Estimation

```python
# two objects
refer1_template_questions = [ "The height of [A] is [Height A], can
    you measure the height of [B]?"]
refer1_template_answers =  ["Since the height of [A] is [Height A], i
    think [B] is [Height B] in height.", ]

refer2_template_questions = ["The width of [A] is [Width A], can you
    measure the width of [B]?"]
refer2_template_answers =  ["Since the width of [A] is [Width A], i
    think the width of [B] is [Width B]"]

refer3_template_questions = ["The height of [A] is [Height A], can you
     measure the size of [B]?"]
refer3_template_answers =  ["Since the height of [A] is [Height A], i
    think the size of [B] is [Length B] x [Width B] x [Height B]."]
```

```
1  # three objects
2  refer4_three_template_questions=["The height of [A] is [Height A],
       what is the height of [B] and [C] ?"]
3  refer4_three_template_answers=["Since the height of [A] is [Height A],
        i think the height of [B] is [Height B] and the height of [C] is
       [Height C]."]
4
5  refer5_three_template_questions=["The distance between [A] and [B] is
       [dis A2B], what is the distance between [B] and [C] ?"]
6  refer5_three_template_answers=["Since the distance between [A] and [B]
        is [dis A2B], i think the distance between [B] and [C] is [dis
       B2C]."]
```

## C   Evaluation Details

### C.1   GPT-4 Evaluation for MSMU-Bench

We resort to LLMs (i.e. GPT-4-Turbo) to evaluate the results. For quantitative queries, GPT-4 extracts numerical values from the responses, and we calculate the success rate using a predefined threshold. The prompt used to extract numerical values is shown as below:

---

You should help me to evaluate the response given the question and the correct answer. You need to convert the measurement of the correct answer and response to meters. The conversion factors are as follows: 1 inch = 0.0254 meters. 1 foot = 0.3048 meters. 1 centimeter (cm) = 0.01 meters. You should output two floats in meters, one for the answer, and one for the response. If the answer or response contains more than one number for prediction, you should output the List that contains the numbers. The output should be in JSON format.

Example 1:
Question: How tall is the long brown table opposite the crossed table?
Answer: The height of the long brown table opposite the crossed table is 1.02 m.
Response: It is 2.17 meters wide.
"answer_in_meters": 1.02, "response_in_meters": 2.17

Example 2:
Question: what's the total number of chairs in the image?
Answer: 2.
Response: There are 2 chairs.
"answer_in_meters": 2,"response_in_meters": 2

Example 3:
Question: What is the size of the dark pillow?
Answer: The dark pillow is with the size of 0.8 m x 0.63 m x 0.55 m
Response: It is 35.9 inches wide.
"answer_in_meters": [0.78,0.63,0.55], "response_in_meters": 0.91

Example 4:
Question: The height of the bed is 0.81 m, what is the height of the table and nightstand?
Answer: Since the height of the bed is 0.81 m, i think the height of the table is 1.02 meters and the height of the nightstand is 0.93 meters.
Response: Since the height of the bed is 0.81 m, i think the height of the table is 1.36 meters and the height of the nightstand is 0.77 meters.
"answer_in_meters": [1.02,0.93], "response_in_meters":[1.36,0.77]

Your Turn:
Question: [Question]
Answer: [Answer]
Response: [Pred]

---

For qualitative questions, GPT-4 scores the model's answers between 0 and 1. The prompt is shown as below:

> You should help me to evaluate the response given the question and the correct answer. To mark a response, you should output a single integer between 0 and 1. 1 means that the response perfectly matches the answer. 0 means that the response is completely different from the answer. The output should be in JSON format.
>
> Example 1:
> Question: Is the blue bed to the left of the curtain from the viewer's perspective?
> Answer: Indeed, the bed is to the left of the curtain.
> Response: Yes, the blue bed is positioned on the left side of the curtain.
> "your_mark": 1
>
> Example 2:
> Question: Between the wooden table and the black chair, which on is taller?
> Answer: The wooden table is taller.
> Response: The chair.
> "your_mark": 0
>
> Example 3:
> Question: What is the tallest among the table, the chair, and the curtain?
> Answer: The tallest is the curtain.
> Response: The curtain.
> "your_mark": 1
>
> Your Turn:
> Question: [Question]
> Answer: [Answer]
> Response: [Response]

## C.2 Q-Spatial

Following the official setting, the evaluation threshold for Q-Spatial is 2.0. And the system prompt used is shown below:

> You will be provided with a question and a 2D image.
> The question involves measuring the precise distance in 3D space through a 2D image.
> You will answer the question by providing a numerical answer.
>
> For example:
> Question: What is the distance between the two chairs?
> Answer: The minimum distance between the two speckled pattern stool chairs is 1 meter.

## C.3 SpatialRGPT

Following the official setting, the evaluation threshold for SpatialRGPT is 1.25. The prompts used to evaluate the qualitative and quantitative questions are the same as those used in MSMU-Bench.

# D  Detailed Results on SpatialRGPT-Bench

More detailed results are shown in Table 7 and Table 8. Our SD-VLM shows the best performance on quantitative tasks such as Height, Vertical Distance, Horizontal Distance, and Direct Distance and qualitative tasks such as Big/Small, Behind/Front, Left/Right, and Tall/Short.

| Model | SpatialRGPT-Bench | | | | |
| | Height | Width | Vertical Distance | Horizontal Distance | Direct Distance |
|---|---|---|---|---|---|
| GPT-4o | 7.8 / 0.76 | 9.0 / 0.67 | 15.1 / 0.61 | 18.0 / 0.65 | 14.9 / 0.64 |
| Gemini-2 | **42.2** / 1.63 | 26.2/ 0.51 | 12.3 / 0.68 | 25.0 / 3.89 | 9.4 / 4.75 |
| Qwen2.5-VL-72B | 31.8 / 1.38 | 23.8/ 0.57 | 8.5 / 0.71 | 8.0 / 0.84 | 9.4 / 0.70 |
| InternVL-3-78B | 38.8 / 2.42 | 23.7 / 0.70 | 24.5 / 1.04 | 17.0 / 0.72 | 13.4 / 0.72 |
| LLaVA-1.5-7B | 31.0 / 1.33 | 24.6 / 0.55 | 7.5 / 0.73 | 10.0 / 0.82 | 7.9 / 0.73 |
| SpatialBot | 28.4 / 2.16 | 20.5 / 0.72 | 6.6 / 0.70 | 8.0 / 3.39 | 2.4 / 0.87 |
| SpatialRGPT | 41.3 / 0.48 | **44.2** / 0.51 | 24.5 / 0.58 | 13.0 / 0.64 | 20.5 / 0.55 |
| Ours | **42.2** / 0.55 | 26.2 / 0.50 | **35.8** / 0.50 | **37.0** / 0.45 | **25.2** / 0.55 |

Table 7: Results on quantitative tasks in SpatialRGPT-Bench. We report the success rate and absolute relative error for SpatialRGPT-Bench.

| Model | SpatialRGPT-Bench | | | | | |
| | Big/Small | Behind/Front | Left/Right | Tall/Short | Wide/Thin | Below/Above |
|---|---|---|---|---|---|---|
| GPT-4o | 55.1 | 59.8 | 63.6 | 56.3 | 55.6 | **72.5** |
| Gemini-2 | 54.1 | 44.6 | 59.1 | 67.7 | 55.6 | 64.2 |
| Qwen2.5-VL-72B | 60.2 | 54.3 | 67.0 | 66.7 | 56.7 | 63.3 |
| InternVL-3-78B | 60.2 | 55.4 | 64.8 | 66.7 | **60.0** | 65.8 |
| LLaVA-1.5-7B | 16.3 | 40.2 | 23.9 | 32.3 | 17.8 | 27.5 |
| SpatialBot | 61.2 | 45.7 | 59.1 | 58.3 | 54.4 | 56.6 |
| SpatialRGPT | 59.2 | 56.5 | 39.8 | 65.6 | 55.6 | 70.0 |
| Ours | **61.2** | **67.4** | **68.2** | **69.8** | 58.9 | 67.5 |

Table 8: Performance of various baselines on the qualitative spatial tasks in SpatialRGPT-Bench.

# E Ablation Study on Estimated Depth

We have conducted a further investigation about the sources of depth maps. From the data presented in the Table 9, it is evident that the use of ground-truth depth maps during both training and inference phases leads to the best performance ($57.71\%$) on the MSMU-Bench dataset. This suggests that the accuracy of depth information is crucial for the model's ability to process and interpret spatial data effectively. If the ground-truth depth is not provided, the overall success rate of the model with estimated depth maps is still competitive. It is noteworthy that a significant disadvantage can be observed when the model is not equipped with any depth information, revealing the importance of incorporating depth priors into the VLM framework.

# F Ablation Study on Normalization

To bridge the gap between different sources of depth maps, we conduct normalization in the depth map before depth positional encoding, which can be formulated as:

$$depth_{norm} = \frac{depth - depth_{min}}{depth_{max} - depth_{min}} * \alpha, \tag{11}$$

where $\alpha$ represents the normalization coefficient, which restricts the maximum value of the depth map.

As shown in Table 10, the highest success rate in MSMU-Bench is achieved when $\alpha$ is 100.

# G Limitations

MSMU concentrates on indoor settings, featuring objects typical of domestic environments, reflecting our source datasets' composition. It narrows the model's applicability to social or dynamic interaction contexts. However, our model still exhibits strong adaptability, as evidenced by its solid performance on benchmarks like SpatialRGPT-Bench, which contains abundant outdoor scenes. In the future, we will explore larger base models and alternative architectures, such as Qwen-VL, to further investigate our proposed framework.

| Training Setting | Inference Setting | MSMU-Bench |
|---|---|---|
| w/ GT depth | w/ GT depth | 57.71 |
| w/ GT depth | w/ estimated depth | 56.31 |
| w/ estimated depth | w/ GT depth | 54.17 |
| w/ estimated depth | w/ estimated depth | 55.35 |
| w/o any depth | w/o any depth | 46.73 |

Table 9: Ablation study on the sources of depth maps.

| $\alpha$ | MSMU-Bench |
|---|---|
| 50 | 49.50 |
| 100 | 56.31 |
| 200 | 53.67 |
| 500 | 52.98 |

Table 10: Ablation study on the normalization coefficient.

## H  Broader Impact

Our model enhances its role as a robust multi-modal generalist by demonstrating superior precision in spatial understanding. This capability is particularly valuable in embodied AI applications, where it aids robots in perceiving their surroundings with greater accuracy and performing precise manipulations. In addition, our model, which is based on large language models, may encounter issues with hallucination, posing significant challenges when deploying the model in real-world environments.

## I  More Result Comparisons on MSMU-Bench

Figure 8 and 9 illustrate more result comparisons on MSMU-Bench tasks. Our model shows a consistent advantage in spatial measuring and understanding. It is worth noting that our model is able to reason about complex spatial tasks with the chain-of-thought while other models fail to answer or return an incorrect answer.

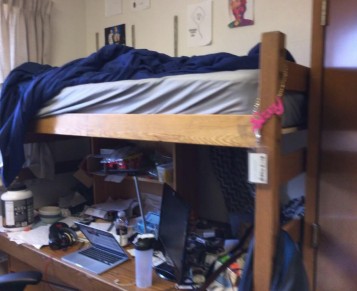

**Question (Reference Object Estimation):**
The height of the right shelf is 1.65 meters, can you measure the height of the desk with laptop and water bottle?
**Answer:**
**GT:** *Since the height of the right shelf is 1.65 meters, i think the desk with laptop and water bottle is 0.71 meters in height*
**GPT-4o:** I'm unable to directly measure distances or heights in images. However, you can possibly estimate the desk height by comparing it to the 1.65-meter tall shelf if both are on the same plane and at a similar distance from the camera.
**LLaVA-1.5:** N/A.
**SpatialRGPT:** The height of the desk with laptop and water bottle is 0.55 meters.

**Ours:** To determine the height of the desk with the laptop and water bottle, I should think it through step by step carefully. Firstly, the right shelf's height is given as 1.65 meters, which can be used as a reference scale. Observing the image, the desk appears to be slightly shorter than the right shelf. By estimating the proportion, the desk seems to be around 50% of the shelf's height. Therefore, multiplying 1.65 meters by 0.50 gives an approximate height of 0.825 meters for the desk. So, the height of the desk with the laptop and water bottle is estimated to be 0.825 meters.

Figure 8: An example of the reference object estimation question in MSMU-Bench.

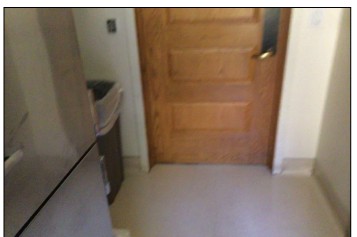

**Question (Existence):**
Can you provide the size measurement of washing machines?
**Answer:**
**GT:** *Can not find washing machines.*
**GPT-4o:** I'm unable to provide size measurements as there are no washing machines visible in the image.
**LLaVA-1.5:** N/A.
**SpatialRGPT:** The size of washing machines can vary depending on the model and brand.
**Ours:** There is no washing machines in the image.

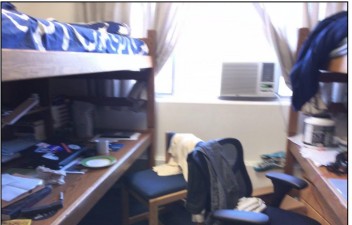

**Question (Counting):**
How many chairs are there in the image?

**Answer:**
**GT:** *2.*
**GPT-4o:** There are two chairs in the image.
**LLaVA-1.5:** There are two chairs in the image.
**SpatialRGPT:** There are 3 chairs in the image.
**Ours:** There are 2 chairs.

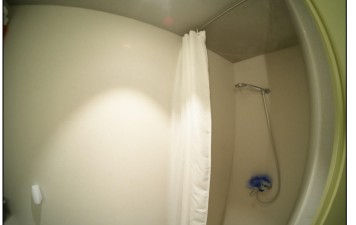

**Question (Grounding):**
What object is located at (0.53,0.59)?

**Answer:**
**GT:** *The shower curtain.*
**GPT-4o:** The object located at (0.53, 0.59) is a showerhead.
**LLaVA-1.5:** The shower curtain is located at (0.53,0.59).
**SpatialRGPT:** The object located at (0.53,0.59) is a shower.
**Ours:** It is the shower curtcain.

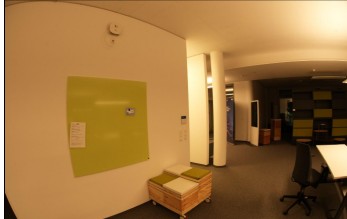

**Question (Scale Estimation):**
What is the height of the office chair?

**Answer:**
**GT:** *The office chair measures 1.08 meters in height.*
**GPT-4o:** The height of the office chair is 0.8 meters.
**LLaVA-1.5:** The office chair is of average height.
**SpatialRGPT:** The height of the office chair is 30 inches.
**Ours:** The height of the office chair is 1.06 m.

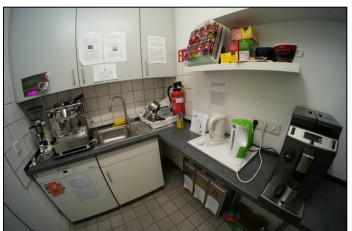

**Question (Relative Position):**
Can you confirm if the coffee machine is positioned to the left of the dishwasher?
**Answer:**
**GT:** *No, the coffee machine is to the right of the dishwasher.*
**GPT-4o:** Yes, the coffee machine is positioned to the left of the dishwasher.
**LLaVA-1.5:** N/A.
**SpatialRGPT:** Yes, the coffee machine is located to the left of the dishwasher.
**Ours:** No, the coffee machine is to the right of the dishwasher.

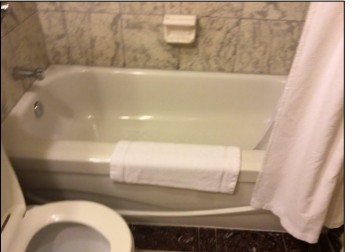

**Question (Scale Comparison):**
Between the white toilet and the shower curtain, which one is larger?
**Answer:**
**GT:** *The size of the white toilet is 0.69 m x 0.64 m x 0.71 m and the shower curtain is 41.29 cm x 30.74 cm x 183.82 cm, so the white toilet is larger.*
**GPT-4o:** In terms of overall size, the shower curtain is larger than the toilet.
**LLaVA-1.5:** N/A.
**SpatialRGPT:** The shower curtain is larger than the toilet.
**Ours:** The size of the white toilet is 0.74 m x 0.49 m x 0.77 m and the shower curtain is 0.44 meters x 0.31 meters x 1.89 meters, so the white toilet is larger.

Figure 9: More results on MSMU-Bench.

