# OpenReview forum: "SD-VLM: Spatial Measuring and Understanding with Depth-Encoded Vision-Language Models"
_NeurIPS.cc/2025/Conference — NeurIPS 2025 poster_

### Official Review · Reviewer_hh1L · 2025-07-01

**Clarity:** 4
**Significance:** 4
**Originality:** 4
**Rating:** 4
**Confidence:** 3

**Summary:**

In this paper, the authors propose a new dataset with precise spatial annotations and introduce depth positional encoding method which can strengthen VLM's spatial awareness. Their experiments and ablations show their method not only achieve SOTA performance on their proposed dataset but also on other spatial understanding bechmarks.

**Questions:**

1. When using LLMs or VLMs, do the authors only prompt once or several times and pick the best answers?

2. When filtering image and selecting objects, how can you define the objects are partially occluded? To be specific, to what extent, the objects are excluded from your dataset. And in semantic disambiguation, do authors try other VLM instead of Qwen2.5-VL to relabel the objects? How can the authors make the Qwen2.5-VL label the object correctly?

3. Can authors provide statistics of your dataset, for example, for each QA pairs, how many object will be involved and their relative sizes.

**Ethical Concerns:**

["NO or VERY MINOR ethics concerns only"]

**Final Justification:**

Most of my main concerns are resolved by the authors' explanations. I will keep my rating (borderline accept).

**Limitations:**

Yes.

**Paper Formatting Concerns:**

No.

**Quality:**

4

**Strengths And Weaknesses:**

Strengths:
1. The topic is interesting.
2. The paper is well-structured and easy to follow.
3. The experiments and ablations are well-designed and comprehensive.
4. Datasets is also well-designed.

Weaknesses:
1. Line 107, $K$ is not defined, if i am not wrong, it should be camera intrinsics.
2. Some details are missing, I will put these in Question Section.

---

> ### Author Rebuttal · Authors · 2025-07-31
>
> Thank you for your constructive reviews and finding our work interesting. We have added further explanations and statistics to address your concern regarding the LLM/VLM usages and our dataset.
>
> **Q1: The definition of the** $\mathbf{K}$
>
> Yes, $\mathbf{K}$ is the camera intrinsics. Thanks for pointing this out. We will add it to the main paper in the revised version.
>
> **Q2: When using LLMs or VLMs, do the authors only prompt once or several times and pick the best answers?**
>
> We utilize different LLMs or VLMs in some parts of data generation pipeline. The main purpose of using LLMs or VLMs is to enrich the text including the detailed object names and chain of thought. The numerical values are directly extracted from 3D scene data so that the accuracy of numerical annotaions can be guaranteed. The final samples are generated by pre-defined templates.
>
> For generating MSMU&#8209;CoT samples, considering the large number of samples, we are unable to manually pick the outputs of large models. Instead, to ensure the data quality,
> we resort to another LLM (DeepSeek V3) to verify the generated reasoning path. We call it "LLM collaboration".
>
> In terms of GPT-4 evaluation, we prompt it once for each question to make a fair comparison.
>
> **Q3: The definition of occluded objects during filtering images**
>
> We calculate the visible size of the mask that are projected to the 2D plane. To be specific, the projected area size of the object can be denoted as $V\_o$and the area size that is visualized in the image is denoted as $V\_{visible}$. We compute the ratio of $V\_{visible} / V\_o$. If the ratio is below the threshold,  the object will be excluded from the image pair.
>
> **Q4: How can the authors make the Qwen2.5-VL label the object correctly?**
>
> We use Qwen2.5&#8209;VL since it is an open-sourced and superior VLM model at the time we constructed the dataset. To make Qwen2.5&#8209;VL label the object correctly, we provide not only the object label in the prompt, but also the cropped object image patch. The attributes (color, texture, position, etc.) of objects are different. The Qwen2.5&#8209;VL is capable to distinguish them and enrich the details of object descriptions. So we did not try other VLMs.
>
> **Q4: More statistics of MSMU dataset**
>
> Thank you for your suggestions. We have already provided some statistics about our dataset in the Figure 2 of main paper, and Figure 1 of supplementary.
> Here are more statistics as follows. For each QA pair, there are about 1.8 objects. As shown in the table below, we can observe that, among all objects: the average width=0.51m with std=0.44m; average height=0.88m with std=0.63m; average distance=1.73m with std=1.18m.
>
> |       |  Mean  | Std |
> |---------------|-----------------|-------------------|
> | Width   | 0.51  | 0.44     |
> | Height   |  0.88  | 0.63    |
> | Distance   |   1.73 | 1.18    |

---

> > ### Comment · Reviewer_hh1L · 2025-08-04
> >
> > Thanks for the rebuttal, my main concerns are resolved.

---

> > > ### Author Response · Authors · 2025-08-06
> > >
> > > Thanks for your kind response and appreciation of our work.

---

### Official Review · Reviewer_vpgs · 2025-07-01

**Clarity:** 3
**Significance:** 2
**Originality:** 3
**Rating:** 4
**Confidence:** 4

**Summary:**

The authors propose to mitigate the challenges of VLM’s with spatial understanding, by developing the dataset, with precise spatial annotations with 2.5 mil numerical annotations, generated from real 3D scenes. They analyze different ways to incorporate depth information, fine-tune LlaVa-7.5B model and carry out comprehensive experiments improving the performance on the main spatial benchmarks.

**Questions:**

It would be help to make more quantitative statements comparing the methods with previous models that try to generate datasets for spatial information and also comment on possible shortcomings of their approach. What are commonly encountered errors of 3D scene graph construction ?  (in comparison with methods that use detectors and segmentation)

More precise definition of the tasks evaluated in Table 1, what are the templates of VQA's ?

The improvement in the grounding performance is quite significant. What is the insight behind this ?

**Ethical Concerns:**

["NO or VERY MINOR ethics concerns only"]

**Final Justification:**

After reading the rebuttal, other reviews and discussion with the authors, I have upgraded my rating to borderline accept.

**Limitations:**

yes

**Paper Formatting Concerns:**

None.

**Quality:**

3

**Strengths And Weaknesses:**

Strengths:

- new spatial quantitative annotation and  new large scale dataset with 2.5 mil numerical annotations, generated from real 3D scenes
- examination how to incorporate the depth information and ablation on different choices
- improved performance on the spatial benchmarks and other tasks

Weaknesses:

Better positioning of the architecture wrt other models (e.g. SpatialVLM's that have slightly different architectures).
Better discussion of the insights that go beyond improvement of performance and depth representation,  as
not surprising that fine-tuning with higher quality dataset yields better results.

---

> ### Author Rebuttal · Authors · 2025-07-31
>
> We thank you for your constructive reviews. We give more explanations on our insights and comparisons with previous methods to address your concerns as follows.
>
> **Q1: Insights of facilitating the spatial capability of VLMs**
>
> Our work provides novel insights into the spatial ability of VLMs from the viewpoint of 3D reconstruction. We assume that the spatial reasoning capability of VLMs is based on its understanding of the 3D spatial structure of an image. As discussed in Section 3 of the main paper, we review the procedure recovering 3D structure from an image. The precise physical measurements and depth are both necessary to reconstruct 3D geometry.
>
> We first construct the dataset with precise physical numerical annotations. However, merely having high&#8209;quality data is not enough, as z&#8209;axis depth information is essential for constructing a comprehensive 3D scene. Then, we introduce Depth Positional Encoding (DPE) to incorporate depth information into image features. This information allows the model to implicitly learn 3D spatial concepts.  As shown in the following table, our model without depth, merely trained on MSMU dataset, falls behind our model equipped with DPE, even behind GPT-4o on Q-Spatial++.
>
> | Model | MSMU-Bench | Q-Spatial++ |
> | --- | --- | --- |
> | GPT-4o | 32.28 | 52.0 |
> | Ours w/o depth | 46.7 | 51.1 |
> | Ours w/ depth | 56.3 | 56.2 |
>
> Furthermore, we explore different ways of integrating depth and finally propose DPE, which can fully unleash the potential of numerical spatial data. Detailed analysis and insights can be referred to **Q1** in response to Reviewer 1Nia.
>
> To further explain the advantage of DPE, we provide more empirical results on different datasets in the following table. Results on three benchmarks show that DPE is a more effective method to unleash the potential of 3D spatial awareness.
>
> | Model | MSMU-Bench | Q-Spatial++ | SpatialRGPT-Bench |
> | --- | --- | --- | --- |
> | Ours w/o depth | 46.7 | 51.1 | 30.1 |
> | depth as image | 22.6 | 46.2 | 25.5 |
> | depth as prompt | 48.8 | 54.4 | 30.7 |
> | depth as token | 35.7 | 40.9 | 23.6 |
> | DPE | 56.3 | 56.2 | 33.3 |
>
> **Q2: Better positioning of the architecture wrt other models**
>
> SpatialVLM inherits the structure of PaLM-E [1] without architecture modification or depth integration. SpatialRGPT inherits the VILA [2]  with independent image/depth linear connectors. Besides, SpatialRGPT uses Region&#8209;feature Extractor (from RegionGPT [3]) to extract features in given bboxes. SpatialBot directly inherits traditional VLM architecture and trains the model to recognize depth values.
>
> Our model uses LLaVA as our backbone. We further integrate depth via a simple yet effective method, DPE, which is also mentioned by reviewer 1Nia. Figure 4 of the main paper illustrates the differences between depth integration methods. DPE is free of introducing extra modules of VLMs like SpatialRGPT or employing a specialized depth api like SpatialBot, and gets better spatial performances at a small cost.
>
>
> **Q3: More comparisons with previous spatial information generating methods**
>
> Compared with previous spatial information generating methods, our data source (ScanNet, ScanNet++) is from reconstructed 3D meshes with physical sizes and manually annotated semantic labels. In ScanNet++, each scene is captured with a high&#8209;end laser scanner at sub&#8209;millimeter resolution, with an average distance of 0.9mm between points in a scan.  Therefore, the numerical measurements to construct the 3D scene graph in our dataset are from high-fidelity 3D geometry.
>
> Previous detector/segmentation-based pipelines may encounter errors in two aspects:
>
> (1) **Recognition errors caused by segmentation/detection models**
> SpatialRGPT uses GroundingDINO [4] to localize objects. The zero&#8209;shot performance of GroundingDINO is only 46.7 AP. SpatialVLM uses ShapeMask[5] whose instance segmentation accuracy is only 40.0 AP. Although these two detectors are currently superior deep learning models, noisy masks will lead to inaccurate object shapes (occluded or including background). Furthermore, the object sizes (length/width/height/) will be noisy.
>
> (2) **Noise from camera calibration and depth estimation models**
> Compared to the millimeter&#8209;level errors of laser scanner, current powerful monocular depth estimation models struggle to achieve centimeter&#8209;level accuracy. SpatialVLM uses ZeoDepth [6] to get the depth map. It's zero&#8209;shot transfer error is at least 5%. Camera calibration models also introduce noise. These noise will lead to inaccurate reconstruction from an image to 3D geometry.
>
> The noise from various deep learning models could be accumulated and influence the precision of numerical annotations if they are combined to reconstruct the 3D scene graph.
>
> **Q4: Definitions of MSMU tasks**
>
> We have given the task categories and definitions in Section 4.1 of the main paper and examples of different tasks in Figure 1 of the main paper. The template details used to generate samples were provided in the Section B.3 in the supplementary.
>
> **Q5: The insight behind grounding performance improvement**
>
> As mentioned in **Q1**, our methods enable the VLM to implicitly understand the corresponding 3D scene of an image. VLMs can better understand the volume occupation and relative distribution of objects in the space. Hence, VLMs can distinguish an object and the background clearly in an image, so that the grounding performance is improved.
>
>
>
> **References**
>
> \[1\] Driess et al. Palm-e: An embodied multimodal language model. ICML, 2023.
>
> \[2\] Lin et al. VILA: On Pre-training for Visual Language Models. CVPR 2024.
>
> \[3\] Guo et al. RegionGPT: Towards Region Understanding Vision Language Model. CVPR 2024.
>
> \[4\] Liu et al. Grounding DINO: Marrying DINO with Grounded Pre-Training for Open-Set Object Detection. ECCV 2024.
>
> \[5\] Kuo et al. ShapeMask: Learning to Segment Novel Objects by Refining Shape Priors. ICCV 2019.
>
> \[6\] Bhat et al. ZoeDepth: Zero-shot Transfer by Combining Relative and Metric Depth.

---

> > ### Comment · Reviewer_vpgs · 2025-08-05
> > **positioning with respect to SpatialVLM**
> >
> > Thank you for the explanation, for highlighting the differences between related architectures, specifically PALM-e used in Spatial VLM. For understanding 3D relationships more significant difference is the nature of pre-training of visual backbone in PALM-e that uses object scene representations and is pre-trained on robotics tasks and does not require explicit incorporation of depth information.  I appreciate your the attempt to enhance the 3D spatial reasoning capabilities on the models that are pre-trained on 2D data. My remaining questions have been addressed in the rebuttal.

---

> > > ### Author Response · Authors · 2025-08-06
> > >
> > > Thank you for your response. As you pointed out, we aim to improve the spatial ability of 2D pre-trained models, since spatial reasoning ability is crucial. We hope our work can push the boundary of current MLLMs.

---

### Official Review · Reviewer_1Nia · 2025-07-03

**Clarity:** 4
**Significance:** 4
**Originality:** 3
**Rating:** 5
**Confidence:** 4

**Summary:**

This paper studies the 3D spatial relationship problem. The author proposes a new dataset called Massive Spatial Measuring and Understanding (MSMU) with precise spatial annotations. The MSMU datasets cover massive quantitative spatial tasks with 700K QA pairs and 2.5M physical numerical annotations. The author benchmarked tens of state-of-the-art multi-modality large language models including GPT model family, Gemini, Qwen, LLaVA, SpatialRGPT, etc. The author also identifies the key issue of the existing vision-language model for 3D spatial understanding and proposes a simple depth positional encoding method to strengthen the vision-language model's spatial awareness. Results show that the proposed method, i.e., SD-VLM, significantly outperforms existing SOTA with a large margin in both the new dataset and existing spatial understanding benchmarks.

**Questions:**

All the questions are fully discussed in the Strengths and Weaknesses section. Please refer to those sections for all the details.

**Ethical Concerns:**

["NO or VERY MINOR ethics concerns only"]

**Final Justification:**

The rebuttal has addressed my concern of the novelty of the depth positional encoding. Overall this is a good paper and I maintain my recommendation for Acceptance

**Limitations:**

Yes. The author has discussed the limitations of their work in their supplementary materials.

**Paper Formatting Concerns:**

N/A.

**Quality:**

4

**Strengths And Weaknesses:**

**Strengths**:

1. Overall, the paper is very well-written and well-motivated. The introduction precisely summarizes the existing practices and challenges of 3D reasoning with vision-language models, and the shortcomings of the existing datasets proposed in SpatialVLM and SpatialRGPT. This provides a strong motivation for proposing a new dataset to push the boundary of the 3D spatial understanding capability of the multi-modal large language model (MLLM), as it is a critical capability for MLLM.

2. The proposed MSMU dataset is extensive and comprehensive. It contains 700K QA pairs with 2.5M numerical annotations, generated from real 3D scenes. The author provides a clear description of the data generation pipeline in Figure 2 and the comparison of the proposed dataset with existing datasets in Figure 3, making it clear to evaluate the significance of the proposed dataset. The MSMU has a large task comprehensiveness compared to existing datasets, which is beneficial to evaluate the MLLM performance on 3D spatial understanding under different aspects. Overall, the reviewer believes that such a dataset will have a large contribution to the community.

3. The idea of depth positional encoding is simple yet very effective. The reviewer appreciates that the author provides a clear comparison of the existing practice of integrating depth information into the MLLM, as shown in Figure 4. This will help the reviewers understand the high-level idea of how the depth information can be integrated into MLLM. The experiments in Section 6 further show the effectiveness of the depth positional encoding idea, where it outperforms tens of state-of-the-art MLLMs in different benchmarks.

4. The experiment section is very well-designed. The author compared tens of existing SOTA MLLMs under different 3D spatial measuring and understanding tasks, which provide many insights and takeaways for the community to further improve the VLMs in the future. Moreover, the results on other existing spatial benchmarks further verify the effectiveness of the proposed SD-VLM as it consistently outperforms SOTA with a large margin.

&nbsp;

**Weaknesses**:

**Technical novelty of the depth positional encoding**: As the positional encoding is well-known in the literature and commonly used in practice, the technical novelty is limited. It would be beneficial if the author could provide more insights and analysis to explain why this design is fundamentally superior to other designs, either theoretically or empirically. This can help further improve the contribution of the present submission.

**Open-source of the dataset and code**: One concern is that the author does not indicate whether the dataset and the code will be publicly available in the future. This will influence the significance of the submission.

&nbsp;

Reference:

---

> ### Author Rebuttal · Authors · 2025-07-31
>
> We sincerely appreciate your insightful feedback. We’re pleased to hear that you found our work well&#8209;motivated and Depth Positional Encoding (DPE) simple yet very effective. We provide more analysis to explain the superiority of DPE.
>
> **Q1: The superiority of DPE**
>
> Theoretically, depth positional embeddings are injected into the query–key score of each attention head, so every image token pair (i,j) receives a distance-aware bias. Specifically, self-attention computes relevance via the dot-product of query ($\mathbf Q$) and key ($\mathbf K$). If depth position encodings are fused additively:
>
> $\mathbf Q=(\mathbf E^{image}+\mathbf E^{depth})\ \cdot \mathbf W\_{Q},\qquad \mathbf K=(\mathbf E^{image}+\mathbf E^{depth})\ \cdot \mathbf W\_{K},$
>
> the dot&#8209;product $QK^T$ naturally decomposes into **image&#8209;image**, **image&#8209;depth**, and **depth&#8209;depth** interaction terms. The model can directly learn the relative importance on depth of each image token.
>
> In contrast, existing depth integration methods can all be categorized as token concatenation. Putting “depth” as a text prompt or a side token only changes the context embeddings in the sequence. This introduces an unavoidable burden that "depth" tokens have to attend to all other "image" or "text" tokens, disturbing the relation between an image patch with its corresponding depth information. Therefore, the problem of **Redundancy of Concatenation** is inevitable.
>
> Empirically, more comparative results in SpatialRGPT&#8209;Bench and Q&#8209;Spatial++ among different depth integration methods are shown below. Our DPE demonstrates a consistent advantage over other methods.
>
> | Model | MSMU-Bench | Q-Spatial++ | SpatialRGPT-Bench |
> | --- | --- | --- | --- |
> | depth as image | 22.6 | 46.2 | 25.5 |
> | depth as prompt | 48.8 | 54.4 | 30.7 |
> | depth as token | 35.7 | 40.9 | 23.6 |
> | DPE | 56.2 | 56.2 | 33.3 |
>
> **Q2: Open-source dataset and code**
>
> As stated in the Abstract, both the dataset and the source code will be released publicly in the future.

---

> > ### Comment · Reviewer_1Nia · 2025-08-04
> >
> > Thank you for the rebuttal and my concerns have been resolved.

---

> > > ### Author Response · Authors · 2025-08-06
> > >
> > > Thanks for your kind response and appreciation of our work.

---

### Official Review · Reviewer_NnRS · 2025-07-03

**Clarity:** 3
**Significance:** 3
**Originality:** 3
**Rating:** 4
**Confidence:** 3

**Summary:**

The authors propose SD-VLM, a framework to enhance the spatial perception abilities. It includes two parts. A dataset/benchmark MSMU with spatial annotations. and a depth-aware positional encoding method. Experiments show that this pipeline could successfully boost the spatial reasoning ability of VLMs.

**Questions:**

1. How robust is the DPE to domain shift and depth estimation noise?
The model uses estimated depth from Depth-Anything-V2. Have the authors evaluated DPE’s performance when using other depth estimators or when applying the model to out-of-domain scenarios (e.g., outdoor scenes or synthetic data)?
2. Does the addition of DPE affect the model’s performance on non-spatial vision-language tasks?
Given that the model is fine-tuned for spatial understanding, it would be important to know whether performance degrades on general VQA benchmarks like GQA or TextVQA that include a broader range of reasoning skills and some other spatial reasoning benchmarks like WhatsUp.
3. I noticed that DPE-sincos has similar performance with DPE-learnable on MSMU-bench. Has this conclusion been evaluated on wider benchmarks? Intuitively, DPE-sincos should be better generalization ability than DPE-learnable.

**Ethical Concerns:**

["NO or VERY MINOR ethics concerns only"]

**Limitations:**

Yes

**Quality:**

3

**Strengths And Weaknesses:**

**Strengths**

1. The model attains strong performance on MSMU and external benchmarks such as Q-Spatial++ and SpatialRGPT-Bench, showing clear gains over prior work.
2. The depth-aware positional embedding is well-motivated and insightful; it is likely to be useful method to boost future VLM performance.
3. I believe the released dataset and benchmark, MSMU, are valuable contributions to the community for advancing spatial reasoning in VLMs and will be beneficial for future research.

**Weaknesses**

All training data are based on ScanNet, as are most of the evaluation settings (on MSMU benchmark in Table 1). Therefore, results on other spatial or general VQA benchmarks—such as WhatsUp, TextVQA, and GQA—would better demonstrate the broader applicability of the proposed method.

---

> ### Author Rebuttal · Authors · 2025-07-31
>
> We appreciate your insightful feedbacks. We’re glad to hear that you found our work valuable and insightful. Further explanations and additional experimental results are added to address your concerns.
>
> **Q1: Robustness to depth estimation noise**
>
> We take more ablation studies on the depth estimation noise on MSMU-Bench. To be specific, we inject zero&#8209;mean Gaussian noise with standard deviations $\delta \in$  {0.1, 0.3, 0.5, 0.7} into the normalized estimated depth maps. As δ increases, the downstream metrics naturally degrade, yet the overall performance remains competitive. The results demonstrate that our DPE is robust to depth perturbations.
>
> | No noise | $\delta$ = 0.1 | $\delta$ = 0.3 | $\delta$ = 0.5 | $\delta$ = 0.7 | w/o depth|
> | --- | --- | --- | --- | --- | --- |
> | 56.3 | 55.1 (-1.2) | 54.0 (-2.3) | 53.3 (-3.0) | 51.4 (-4.9) | 46.7 (-9.6)  |
>
> **Q2: Using other depth estimators**
>
> We conduct an additional ablation study on the depth estimation backbone by replacing DepthAnything with another powerful depth estimator, UniDepth [1]. As summarized in the table below, our model maintains comparable and competitive performance across various spatial benchmarks. This robustness indicates that it has learned generalizable depth priors rather than overfitting to any specific depth-estimation architecture.
>
> |  | MSMU-Bench | Q-spatial++ | SpatialRGPT-Bench | Average |
> | --- | --- | --- | --- | --- |
> | SpatialRGPT | 29.0 | 43.5 | 28.7 | 33.7 |
> | Ours (UniDepth) | 56.2 | 54.7 | 32.0 | 47.6 |
> | Ours (DepthAnything) | 56.3 | 56.2 | 33.3 | 48.6 |
>
> **Q3: Performance in out-of-domain scenarios**
>
> In terms of the **domain shift** of  **out&#8209;of&#8209;domain scenarios**,  we have conducted evaluation on SpatialRGPT&#8209;Bench which covers a variety of scene data, including nuScenes ( outdoor urban scenes ), KITTI ( outdoor urban scenes ), and Hypersim ( synthetic scenes ). We have achieved the best performance on this benchmark, as shown in Table 2 of the main paper, revealing the generalization ability of our proposed SD-VLM.
>
>
> **Q4: Results on other spatial or general VQA benchmarks**
>
> We conduct experiments on other tasks, including Whatsup, TextVQA, and GQA. Related results are shown in the following table. Our variants perform similarly to the baselines on these benchmarks. It means that the spatial capabilities of VLMs can be enhanced without compromising general VQA performance.
>
> | Model |  Whatsup | TextVQA | GQA |
> | --- |  --- | --- | --- |
> | LLaVA-1.5-7B | 58.3 | 58.2 | 61.9 |
> | Ours  | 60.5 | 55.8 | 59.1 |
>
> **Q5: DPE-sincos vs DPE-learnable on wider benchmarks**
>
> Thank you for your insightful suggestions. To further validate the generalization ability of depth&#8209;encoding strategies, we expanded our ablation to compare sincos and learnable variants.  The table shows that sincos consistently generalizes better. We attribute this to two factors:
>
> (1) learnable embeddings can overfit the specific domain of the training data, limiting robustness when test distributions shift;
>
> (2) sincos encoding is theoretically fixed and frequency&#8209;based, providing a stable, domain&#8209;agnostic prior that transfers reliably to unseen scenes.
>
> | Model | MSMU-Bench | Q-spatial++ | SpatialRGPT-Bench | Whatsup | TextVQA | GQA |
> | --- | --- | --- | --- | --- | --- | --- |
> | DPE-learnable | 56.2 | 50.7 | 25.7 | 56.4 | 51.4 | 56.4 |
> | DPE-sincos | 56.3 | 56.2 | 33.3 | 60.5 | 55.8 | 59.1 |
>
> **References**
>
> [1] Piccinelli et al., UniDepth: Universal Monocular Metric Depth Estimation. CVPR 2024.

---

> > ### Comment · Reviewer_NnRS · 2025-08-04
> >
> > Thank you for sharing the additional results. While most of the new numbers are encouraging, the performance on TextVQA and GQA drops noticeably. Could the authors elaborate on the possible reasons for this decline? Given the gains on other spatial-reasoning benchmarks, it would be helpful to understand why similar improvements do not extend to TextVQA and GQA.

---

> ### Author Response · Authors · 2025-08-06
>
> Thanks for your suggestion. We assume it is because our model trained on MSMU is likely shifted a little toward spatial related domain. For a fair comparison with the baseline LLaVA-v1.5, we conduct a contrast experiment - integrating LLaVA's general training data (LLaVA-v1.5-mix665k) into our comprehensive training suite.
>
> | Model | MSMU-Bench | TextVQA | GQA |
> | --- | --- | --- | --- |
> | LLaVA-v1.5-7B (training with general data) | 19.5 | 58.2 | 61.9 |
> | Ours (training with MSMU + general data) | 55.8 | 57.5 | 62.9 |
>
> The results in the table show that the model's overall proficiency on general benchmarks closely mirrors that of the original model.
>
> - The drop in TextVQA still exists but is very small, which can be attributed to the domain gap, as TextVQA is an OCR dataset, centered around detailed text recognition.
>
> - The performance on GQA even surpasses the original backbone. This improvement is likely due to the fact that images in GQA are from daily scenarios, and questions rely to some extent on spatial abilities, which can be enhanced by our MSMU and DPE, e.g., we provide two samples from GQA in the table below.
>
> | Question | Answer |
> | --- | --- |
> | Are there any women to the left of the brown bag? | Yes, there is a woman to the left of the bag. |
> | Is there a pepper to the left of the white food? | Yes, there are peppers to the left of the cheese. |
>
> This suggests that incorporating a broader range of data during training can more effectively sustain the model's foundational skills.

---

### Decision · Program_Chairs · 2025-09-17

**Decision:**

Accept (poster)

**Comment:**

Summary:
This paper introduces SD-VLM, a lightweight depth-encoded VLM trained on the new MSMU dataset for quantitative spatial understanding. It adds DPE to image tokens and reports clear gains on MSMU-Bench, Q-Spatial++, and SpatialRGPT-Bench.

Strengths:
Practical, plug-and-play design (depth→PE added to visual tokens) with consistent improvements across spatial benchmarks; MSMU is a sizable, useful resource. The authors' rebuttal adds robustness evidence (noise injection, swapping depth estimators) and cross-domain results supporting generalization.

Weaknesses:
Reviewers pointed out that the novelty is limited (essentially dataset + standard SFT with simple DPE), and some reviewers question broader generalization beyond ScanNet-style data; mixing with general VQA data shows small trade-offs.

Overall, this is a solid, deployment-friendly work, with clear empirical value. The ACs agree on acceptance.